# Biogeography of *Stigmaphyllon* (Malpighiaceae) and a Meta-Analysis of Vascular Plant Lineages Diversified in the Brazilian Atlantic Rainforests Point to the Late Eocene Origins of This Megadiverse Biome

**DOI:** 10.3390/plants9111569

**Published:** 2020-11-13

**Authors:** Rafael Felipe de Almeida, Cássio van den Berg

**Affiliations:** 1Departamento de Ciências Biológicas, Universidade Estadual de Feira de Santana, Molecular Biology of Plants and Fungi Lab (LAMOL), Av. Transnordestina s/n, Novo Horizonte, Feira de Santana 44036-900, Bahia, Brazil; vcassio@gmail.com; 2Scientifik Consultoria, Petrópolis, Rio de Janeiro 25651-090, Brazil

**Keywords:** ancestral area reconstruction, Atlantic rainforest, time-calibrated phylogenies, Malpighiales, neotropical flora

## Abstract

We investigated the biogeography of *Stigmaphyllon*, the second-largest lianescent genus of Malpighiaceae, as a model genus to reconstruct the age and biogeographic history of the Brazilian Atlantic Rainforest (BAF). Few studies to date have focused on the tertiary diversification of plant lineages in the BAFs, especially on *Stigmaphyllon*. Phylogenetic relationships for 24 species of *Stigmaphyllon* (18 ssp. From the Atlantic forest (out of 31 spp.), three spp. from the Amazon Rainforest, two spp. from the Caatinga biome, and a single species from the Cerrado biome) were inferred based on one nuclear DNA (PHYC) and two ribosomal DNA (ETS, ITS) regions using parsimony and Bayesian methods. A time-calibrated phylogenetic tree for ancestral area reconstructions was additionally generated, coupled with a meta-analysis of vascular plant lineages diversified in the BAFs. Our results show that: (1) *Stigmaphyllon* is monophyletic, but its subgenera are paraphyletic; (2) the most recent common ancestor of *Stigmaphyllon* originated in the Brazilian Atlantic Rainforest/Caatinga region in Northeastern Brazil ca. 26.0 Mya; (3) the genus colonized the Amazon Rainforest at two different times (ca. 22.0 and 6.0 Mya), the Caatinga biome at least four other times (ca. 14.0, 9.0, 7.0, and 1.0 Mya), the Cerrado biome a single time (ca. 15.0 Mya), and the Southern Atlantic Rainforests five times (from 26.0 to 9.0 Mya); (4) a history of at least seven expansion events connecting the Brazilian Atlantic Rainforest to other biomes from 26.0 to 9.0 Mya, and (5) a single dispersion event from South America to Southeastern Asia and Oceania at 22.0 Mya via Antarctica was proposed. Compared to a meta-analysis of time-calibrated phylogenies for 64 lineages of vascular plants diversified in the Brazilian Atlantic Rainforests, our results point to a late Eocene origin for this megadiverse biome.

## 1. Introduction

Malpighiaceae Juss. is one of the most diverse plant families of Neotropical shrubs, trees, and lianas [1], with most species confined to this region [2]. Its species are easily recognized by their remarkable floral conservatism, with flowers frequently bearing a pair of oil-secreting glands at the base of sepals, petals clawed at the base, and a posterior petal differentiated from the remaining lateral four [1,2]. This family has received broad phylogenetic attention in the past few years [2,3,4,5,6,7,8], including more focused investigations on generic delimitations and the phylogenetic position of Old-World clades [2,6]. However, there have been few efforts to determine finer-scale patterns of molecular analyses and historical biogeography in Neotropical Malpighiaceae [4,6,8,9].

*Stigmaphyllon* A.Juss. is one of the several genera in Malpighiaceae re-circumscribed due to recent molecular phylogenetic studies [2,6]. It is currently the second-largest lianescent genus in Malpighiaceae and the only occurring in forested habitats in the tropics and subtropics of America, Africa, Asia, and Oceania [10]. Most species are woody lianas with long-petioled, elliptical to cordate leaves, corymbs or umbels of yellow flowers arranged in dichasia, styles holding laterally expanded appendages in the apex, and schizocarpic fruits splitting into three-winged mericarps, bearing a sizeable dorsal wing (Figure 1 a–h) [10,11]. This genus is currently divided into two subgenera, *Stigmaphyllon* and *Ryssopterys*, treated as separate genera before molecular phylogenies [10]. *Stigmaphyllon* subgenus *Stigmaphyllon* includes ca. 90 species restricted to the Neotropics, except for *Stigmaphyllon bannisterioides* A.Juss., which is also found in West Africa [11]. On the other hand, *S.* subgenus *Ryssopterys* includes ca. 20 species restricted to Southeast Asia and Oceania [10]. The monophyly of *Stigmaphyllon* and its subgenera has been suggested by previous phylogenetic studies but never adequately corroborated because its type species was never sampled [2,3,5,6].

Previous phylogenetic studies for Malpighiaceae suggested a Brazilian Atlantic Rainforest (herein treated as BAF) origin for *Stigmaphyllon*, with *S. paralias* A.Juss. being consistently recovered in several studies as the first lineage to diverge in the genus [2,6]. Even though the BAFs currently comprise more than 15,000 known vascular plant species [12], they are regarded as one of the most threatened hotspots for biodiversity worldwide due to being mostly fragmented and disturbed by most of Brazil’s human population and economic activity [13]. Several biogeographic hypotheses have been proposed to explain the origins of the great biodiversity of the BAFs, such as: 1. Miocene to Pleistocene forest corridors between BAFs and the Amazon rainforests via Cerrado’s gallery forests and/or via the coastal region in Northeastern Brazil [14]; and 2. Pleistocene refugia hypothesis, which suggested that Pleistocene climatic fluctuations led to rainforest fragmentation and promoted divergence of lineages or species in isolated forest fragments or refugia [15]. Pleistocene diversifications in BAFs have been greatly criticized due to the lack of concordance with empirical phylogenetic data, as well as by the evidence that shifts in forest species distribution, rather than fragmentation, have been the main consequences of global glaciations in the Neotropics [16]. Although most previous studies have focused on explaining BAFs biodiversity through Miocene to Pleistocene climatic/geological events, no study to date has focused on timing the age of vascular plant lineages diversification in BAFs.

In this study, we focus on timing the biogeographic history of BAFs using *Stigmaphyllon* as a model genus, supplemented by a meta-analysis of vascular plant lineages diversified in this biome to infer the age of BAFs. More specifically, we: (1) test the monophyly of *Stigmaphyllon* and its subgenera; (2) time-calibrate the phylogenetic tree; (3) estimate the ancestral areas of *Stigmaphyllon*; and (4) shed some light on the age and biogeographic history of BAFs using *Stigmaphyllon* and a meta-analysis of vascular plant lineages diversified in this biome.

## 2. Results

### 2.1. Phylogenetic Analysis

Our combined dataset for ribosomal (ETS, ITS) and nuclear (PHYC) markers contains a total of 2283 characters, of which 1724 characters are constant, 257 characters are variable, but parsimony uninformative, and 302 characters are parsimony informative. The combined analysis of nuclear and ribosomal markers provides higher support for more clades than those based on independent nuclear and ribosomal datasets. Overlapping peaks (paralogous copies) for ETS and ITS were not recorded during electrophoresis and sequencing. The heuristic search for the combined dataset found 15 trees (consistency index, CI = 0.80, retention index, RI = 0.70), and the strict consensus tree includes 17 moderately supported clades (bootstrap percentage, BP > 75; Figure 1). The Bayesian analysis recovered 25 well-supported to moderately supported clades (posterior probabilities, PP > 0.95 and >0.80, respectively; Figure 1).

The monophyly of *Stigmaphyllon* (Figure 1) is strongly supported by Maximum Parsimony (MP) and Bayesian Inference (BI) analyses, being recovered as sister to *Diplopterys*. The clade *Stigmaphyllon* + *Diplopterys* is well supported (BS100/PP1.0) as sister to *Bronwenia* (Figure 1). Within *Stigmaphyllon*, *Stigmaphyllon* subg. *Stigmaphyllon* is recovered as polyphyletic, with its representatives placed in three separate lineages (Figure 1, black bars), named: the early-diverging *Stigmaphyllon paralias* group (Figure 1, dark blue bar), *Stigmaphyllon ciliatum* group (Figure 1, yellow bar) and core *Stigmaphyllon* (here represented by the common ancestor that the lineage *S. fynlayanum* + *S. puberulum* share with the remaining species (Figure 1, black bar)). *Stigmaphyllon* subg. *Ryssopterys* (Figure 1, dark green bar) is recovered as monophyletic in our analyses, being sister to the *Stigmaphyllon ciliatum* group (Figure 1, white bar).

Within core *Stigmaphyllon*, seven major lineages are recovered, here designated as 1. *S. puberulum* group, 2. *S. auriculatum* group, 3. *S. urenifolium* group, 4. *S. lalandianum* group, 5. *S. sinuatum* group, 6. *S. blanchetii* group, and 7. *S. gayanum* group. The *S. puberulum* group (Figure 1, orange bar) is well-supported (BS100/PP1.0), comprising only two species, *S. finlayanum* A.Juss. and *S. puberulum* Griseb., while the *S. auriculatum* group (Figure 1, purple bar) is moderate to well-supported (B77/PP1.0), comprising three species, *S. angustilobum* A.Juss., *S. arenicola* C.E.Anderson, and *S. auriculatum* (Cav.) A.Juss. The *S. urenifolium* group (Figure 1, light green bar) is weak to well-supported (BS70/PP1.0), represented by only its naming species. In contrast, the *S. lalandianum* group (Figure 1, light blue bar) is moderate to well-supported (BS82/PP1.0), comprising four species, S. *alternifolium* A.Juss., *S. bonariense* (Hook. and Arn.) C.E.Anderson, *S. lalandianum* A.Juss., and *S. vitifolium* A.Juss. The *S. sinuatum* group (Figure 1, red bar) is well-supported (BS100/PP1.0), represented by only two species, *S. lindenianum* A.Juss. and *S. sinuatum* (DC.) A.Juss., while the *S. blanchetii* group (Figure 1, rose bar) is weakly-supported (BS-/PP0.60), comprising four species, *S. blanchetii* C.E.Anderson, *S. caatingicola* R.F.Almeida and Amorim, *S. jatrophifolium* A.Juss., and *S. salzmannii* A.Juss. Finally, the *S. gayanum* group (Figure 1, gray bar) is weakly to well-supported (BS-/PP1.0), and represented by four species, *S. cavernulosum* C.E.Anderson, *S. gayanum* A.Juss., *S. macropodum* A.Juss., and *S. saxicola* C.E.Anderson.

### 2.2. Ancestral Area Reconstruction and Divergence Times Estimation

The S-DEC reconstruction suggests that the most recent common ancestor (MRCA) of *Stigmaphyllon* was widespread in the northern portion of the BAFs ca. 26.0 Mya (Figure 2 and Figure 3, node 3, Table 1). A dispersal event (node 3) took place then, splitting the MRCA of the *Stigmaphyllon paralias* group from the remaining species, which dispersed from Seasonally Dry Tropical Forests (SDTFs) to Campos Rupestres in northeastern Brazil ca. 14.0 Mya (Figure 2 and Figure 3, node 4, Table 1). The MRCA of the *S. ciliatum* and *S. timoriense* (DC.) C.E.Anderson clade diverged ca. 22.0 Mya (node 5) and split into those lineages ca. 10.0 Mya (node 6), colonizing dunes vegetation on the Atlantic and Australasian rainforests (Figure 2 and Figure 3, node 6, Table 1). The MRCA of core *Stigmaphyllon* dispersed to the southern portion of the BAFs ca. 19.0 Mya (Figure 2 and Figure 3, node 7, Table 1). The MRCA of the *S. puberulum* group arose 6.25 Mya, being widespread over both the Amazon and Atlantic rainforests, with a vicariant event giving rise to its current lineages (Figure 2 and Figure 3, node 8, Table 1). The MRCA of the remaining species (node 9) remained distributed in the BAFs ca. 17.0 Mya, giving rise to the MRCA of the *S. auriculatum* group ca. 9.4 Mya (node 10). A dispersal event followed by the colonization of dunes vegetation in Southern Brazil gave rise to the *S. arenicola* lineage (Figure 2 and Figure 3, node 10, Table 1). In the Pleistocene, ca. 0.72 Mya, the MRCA of the *S. auriculatum* and *S. angustilobum* arose in the BAFs from Southeastern Brazil, giving rise to those lineages via a dispersal event from rainforests to SDTFs (Figure 2 and Figure 3, node 11, Table 1). The MRCA of *S. urenifolium* (Figure 2 and Figure 3, node 12, Table 1) arose ca. 15.0 Mya via a dispersal and a vicariant event from the BAFs to the Cerrado. The MRCA of the remaining species (Figure 2 and Figure 3, node 13, Table 1) arose ca. 13.0 Mya in the BAFs and, probably, in the Amazon forest, as well. The MRCA of node 14 remained distributed within the Southeastern portion of the BAFs ca. 7.0 Mya (Figure 2 and Figure 3, node 14, Table 1) and diversified into its four main lineages ca. 4.0 Mya (Figure 2 and Figure 3, nodes 15–16, Table 1). Both a dispersal and a vicariant event took place ca. 12.0 Mya giving rise to the MRCA of node 17, which was distributed within the BAFs, Caatinga, and the Amazon rainforest. The lineage of node 18 colonized the Amazon rainforest for the first time, but only diversified ca. 0.5 Mya (Figure 2 and Figure 3, node 18, Table 1).

### 2.3. Meta-Analysis

We identified 113 genera of ferns/lycophytes, gymnosperms, magnoliids, monocots, and eudicots comprising lineages exclusively diversified in the BAF biome (out of a total of 2224 genera currently recorded by the Flora do Brasil Project; Table 2). Only 64 of those genera have estimated diversification ages available from 34 phylogenetic studies published from 2004 to 2020 (Figure 4, Table 2). Most of those studies were published from 2015 to 2020 when molecular clocks and time-calibrated trees were already widespread in phylogenetic literature.

## 3. Discussion

### 3.1. Phylogenetics of Stigmaphyllon

The topology recovered from the combined dataset (i.e., ribosomal + nuclear markers) evidenced that the subgenera of *Stigmaphyllon* proposed by Anderson [10] are paraphyletic. All previous phylogenetic studies of Malpighiaceae sampled mostly Mesoamerican and Amazonian species of the genus [2,3,5,9,44], making it difficult to properly test the monophyly of its subgenera. The only BAF species of *Stigmaphyllon* sampled in previous studies were *S. ciliatum* and *S. paralias*. However, in all these studies *S. paralias* (a Brazilian Atlantic Rainforest lineage) is consistently recovered as sister to all remaining lineages of *Stigmaphyllon*. Additionally, this is the first time *S. auriculatum*, the type species of the genus, is included in a phylogenetic study. On the other hand, our results highly corroborate the previous topologies recovered for *Stigmaphyllon*, with the *S. paralias* group recovered as the first lineage to diverge, followed by the clade comprising the *S. ciliatum* group + *S.* subg. *Ryssopterys* sister to a large clade consisting of species from the *S. tomentosum* group (core *Stigmaphyllon*) [2,9]. Additional species sampling allied to a thorough morphological study on a phylogenetic perspective in *Stigmaphyllon* is urgently required to shed some light on its infrageneric classification.

### 3.2. Divergence Times and Biogeography of Stigmaphyllon

Our divergence times estimation for the MRCA of *Stigmaphyllon* (ca. 26.5 Mya) is similar to the age of 22.5 Mya estimated by Davis et al. [44] and of 21.0 Mya estimated by Willis et al. [9]. The estimated ages for the MRCAs obtained by us for the *S. paralias* group and the *S. ciliatum* + *S.* subg. *Ryssopterys* clade strongly corroborates those previously reported by Willis et al. [9] (17.0, 15.0, and 10.0 Mya, respectively). The only exception was the age of 19.0 Mya estimated by us for core-*Stigmaphyllon* when Willis et al. [9] recovered the estimation of 10.0 Mya for this clade. This apparent difference in age estimates might be due to the more comprehensive sampling of this clade in the present study.

Over these last 26.0 Mya, the lineages of *Stigmaphyllon* went through 13 dispersals and four vicariance events (VE). Worth noting is that VEs in this genus were always followed by a biome shifting event (BSE) from the BAFs to another tropical biome. The first VE followed by a BSE in *Stigmaphyllon* occurred ca. 15.0 Mya in the MRCA of *S. urenifolium*, from the BAFs to the Brazilian Cerrado. Several plant lineages present the same diversification pattern with older ancestors arising in the BAFs and colonizing the Cerrado biome ca. 15 Mya, such as clade 5 of *Amphilophium* Kunth (Bignoniaceae) [52], *Astraea* Klotzsch (Euphorbiaceae) [38], *Dolichandra* Cham. (Bignoniaceae) [33], *Fridericia* Mart. (Bignoniaceae) [52], and *Xylophragma* Sprague (Bignoniaceae) [53]. Dispersal events from forest to open habitats is one of the main factors explaining richness in Neotropical biodiversity [16]. Even though in *Stigmaphyllon,* those dispersal events occurred mostly among forested biomes, its single Cerrado lineage seems to point to an older diversification of this biome, such as in those from Vochysiaceae (20.0–15.0 Mya) [51]. Few species of *Stigmaphyllon* successfully colonized the Cerrado biome, such as *S. jobertii*, *S. occidentale*, *S. tomentosum,* and *S. urenifolium*. Future studies sampling those species in the molecular phylogeny of *Stigmaphyllon* are crucial to test if the genus colonized the Cerrado biome more than a single time. However, several studies seem to present the same pattern pointed by us of forest lineages occupying and diversifying in the Cerrado biome [16], with the opposite rarely recorded in biogeographic studies. We hypothesize that Cerrado lineages colonizing forested biomes might be rare due to the recent diversification of several plant lineages of Neotropical savannas [54].

The second VE followed by a BSE in *Stigmaphyllon* occurred ca. 12.0 Mya in the MRCA of *S. sinuatum* group, from the BAFs to the Amazon rainforest. The same pattern of diversification was recorded in other plant lineages, with Atlantic rainforest MRCAs colonizing the Amazon rainforest at this time, such as *Eugenia* L. clade G [55]. At least 50 dispersal events occurred between the Atlantic and Amazon rainforests [16]. The number of lineage exchanges between these biomes fluctuated over the last 60.0 Mya, with its previous increase starting ca. 12.0 Mya and peaking ca. 6.0–3.0 Mya [16], corroborating our results with *Stigmaphyllon*. The MRCA of *S. finlayanum* group was the fourth VE, followed by a BSE occurring in the genus, ca. 6.30 Mya, from the BAFs to the Amazon rainforest. The same pattern of diversification was recorded in other plant lineages, with Atlantic rainforest MRCAs colonizing the Amazon rainforest at this time, such as the clade 3 of *Amphilophium* Kunth (Bignoniaceae) [52] and the abovementioned study by Antonelli et al. [16].

The third VE followed by a BSE in *Stigmaphyllon* occurred ca. 10.0 Mya in the MRCA of *S.* subg. *Ryssopterys* + *S. ciliatum* group, from the BAFs to the Asian rainforests. This lineage diversified ca. 22.0 Mya, so we hypothesize that its MRCA was widely distributed among dunes vegetation in the Atlantic and Asian rainforests via the Antarctic route. Antarctica’s glaciation process started ca. 34.0 Mya, during the Eocene/Oligocene transition, on high altitude regions of this continent [56]. From 34.0–12.0 Mya, Antarctica had intermittent ice sheet coverage, leaving intact Tertiary pockets of pre-glaciation fauna and flora that only went completely extinct ca. 12.0 Mya with the formation of permanent ice sheets in this continent [57], the same confidence interval recovered in our study for the MRCA of *S.* subg. *Ryssopterys* + *S. ciliatum* group. The same pattern of vicariance event is observed in the *Drymophila* (Australian) + *Luzuriaga* (South American) clade (Alstromeriaceae), in which its MRCA split ca. 23.0 Mya, giving rise to these genera from ca. 10.0 to 4.0 Mya [58].

On the other hand, 14 dispersal events (DE) occurred within *Stigmaphyllon*, mostly within the North-South corridors of the BAFs, and at least three DEs from the BAFS to the Caatinga biome. From these latter three DEs, the first occurred ca. 14.0 Mya in the MRCA of *S. paralias* group, the second occurred ca. 9.0 Mya in the MRCA of *S. saxicola* group and the third occurred ca. 1.0 Mya in the MRCA of the *S. auriculatum* group. From the remaining DEs that happened within the BAFs, at least five of them occurred from North to South ca. 22.0, 17.0, 12.0, 11.0, and 9.0 Mya. The same pattern with Northern BAF lineages colonizing Southern portions of this biome is observed in *Aechmea* (Bromeliaceae) [59]. However, the North-South dispersals in this genus started only ca. 6.0 Mya (Bromeliaceae) [59], being much younger than those in *Stigmaphyllon*.

### 3.3. Time and Diversification of the Atlantic Rainforest

Several implications for understanding the BAFs historical biogeography might be postulated from the biogeographical study of *Stigmaphyllon*. Our results suggest a late-Eocene origin for these forests, that seems to be partially corroborated by the literature. Until the late Paleocene and early Eocene, the Earth’s climate was mostly warmer and more humid than today, suggesting that South America was probably covered by continuous rainforests [60,61]. When comparing the mean ages recovered in published studies for several lineages of ferns, gymnosperms, and angiosperms diversified in the BAFs, we bring new evidence that vascular plants started colonizing these forests over the last 60.0 Mya (Figure 4; Table 2), corroborating the abovementioned authors. The oldest lineage to occupy the BAFs might have been the genus *Barnebya* W.R.Anderson and B.Gates ca. 60 Mya (Malpighiaceae, Eudicots), with the diversification of most Eudicot lineages in these forests occurring from 40.0 to 15.0 Mya (Figure 4; Table 2). During the late Eocene and Oligocene, global episodes of cooling and dryness favored the expansion of grasslands in the southern and central regions of the continent [60,62], which culminated in the formation of a diagonal belt of more open and drier biomes (also known as “dry diagonal”) [63]. The formation of the dry diagonal marked the formation of the Atlantic forest in the east and Amazonia in the west [64]. On the other hand, the colonization and diversification of Magnoliid lineages took place from 18.0 to 3.0 Mya (Figure 4; Table 2), followed by gymnosperm lineages that diversified from 15.0 to 11.0 Mya, and finally from monocot lineages diversifying from 12.0 to 3.0 Mya in these rainforests (Figure 4; Table 2). Fossil records and paleoclimate studies suggest that the BAFs and the Amazon rainforests were re-connected multiple times in the Miocene and Pliocene [64]. Mean ages regarding the colonization and diversification of ferns in the BAFs are still incipient, with a single study on the family Cyatheaceae evidencing its initial diversification in these forests at 30.0 Mya (Figure 4; Table 2). Additionally, from the 113 BAF lineages presented in Table 2, only 64 show mean ages based on time-calibrated phylogenies available in the literature. At least 18 families of angiosperms (Acanthaceae, Amaryllidaceae, Apocynaceae, Araceae, Asparagaceae, Asteraceae, Bignoniaceae, Erythroxylaceae, Fabaceae, Gentianaceae, Lauraceae, Marantaceae, Moraceae, Orchidaceae, Poaceae, Rubiaceae, Rutaceae, and Sapindaceae) with lineages diversified in the Atlantic forest still lack published time-calibrated phylogenies.

Another worth mentioning factor that might have played an important role in the diversification of the BAFs was the uplift of Serra do Mar and Serra da Mantiqueira Mountain Ranges in Eastern Brazil. Those mountains were previously thought to have uplifted around 120.0 Mya, but geological studies from the past decade have pointed to an earlier uplifting age for these mountains, from 60.0 to 30.0 Mya [65]. The tertiary uplift age of these mountains in Eastern Brazil was a direct result of the Andean uplift, coinciding with our results for the colonization of the BAFs by 64 lineages of vascular plants [65].

## 4. Material and Methods

### 4.1. Taxon Sampling and Plant Material

We sampled 24 species of *Stigmaphyllon* (ca. ¼ of the Neotropics’ genus diversity: 18 spp. from the Brazilian Atlantic Rainforest (out of 31 spp.), three spp. from the Amazon Rainforest, two spp. from the Caatinga, and one spp. from the Cerrado biomes), including outgroups *Bronwenia* W.R.Anderson and C.C.Davis and *Diplopterys* A.Juss. From this total, 23 species represent *S* subg. *Stigmaphyllon* and a single species [*Stigmaphyllon timoriense* (DC) C.E.Anderson] represents *S.* subg. *Ryssopterys* (Table 3). For DNA extraction, we used mainly field-prepared silica dried leaves (12–80 mg) and herbarium specimens as necessary (Table 3).

### 4.2. Molecular Protocols

Genomic DNA was extracted using the 2 × CTAB protocol, modified from Doyle and Doyle [66]. Three DNA regions (nuclear PHYC gene, and the ribosomal external and internal transcribed spacers (ETS and ITS)) were selected based on their variability in previous Malpighiaceae studies [2,5,6,8,67]. Protocols to amplify and sequence ETS and ITS followed Almeida et al. [8]. For amplification, we used the TopTaq (Qiagen) mix following the manufacturer’s standard protocol, with the addition of betaine (1.0 M final concentration) and 2% DMSO for the ETS region. PCR products were purified using PEG (polyethylene glycol) 11% and sequenced directly with the same primers used for PCR amplification. Sequence electropherograms were produced on an automatic sequencer (ABI 3130XL genetic analyzer) using the Big Dye Terminator 3.1 kit (Applied Biosystems). Additional sequences for PHYC were retrieved from GenBank (Table 3). Newly generated sequences were edited using the Geneious software [68], and all datasets were aligned using Muscle [69], with subsequent adjustments in the preliminary matrices made by eye. The complete data matrices are available at TreeBase (accession number S21218). This study was authorized by the Genetic Heritage and Associated Traditional Knowledge Management National System of Brazil (SISGEN #A3B8F19).

### 4.3. Phylogenetic Analysis

Analyses were rooted in *Bronwenia,* according to Davis and Anderson [2]. Individual analyses for each marker were performed, and since no significant incongruencies were found, analyses of combined matrices (i.e., nuclear + ribosomal markers) were performed using maximum parsimony (MP) conducted with PAUP 4.0b10a [70]. A heuristic search was performed using TBR swapping (tree-bisection reconnection), and 1000 random taxon-addition sequence replicates with TBR swapping limited to 15 trees per replicate to prevent extensive searches (swapping) in suboptimal islands, followed by TBR in the resulting trees with a limit of 1000 trees. In all analyses, the characters were equally weighted and unordered [71]. Relative support for individual nodes was assessed using non-parametric bootstrapping [72], with 1000 bootstrap pseudoreplicates, TBR swapping, simple taxon addition, and a limit of 15 trees per replicate.

For the model-based approach, we selected the model GTR + I + G using hierarchical likelihood ratio tests (HLRT) on J Modeltest 2 [73]. A Bayesian analysis (BA) was conducted with mixed models and unlinked parameters, using MrBayes 3.1.2 [74]. The Markov chain Monte Carlo (MCMC) analysis was performed using two simultaneous independent runs with four chains each (one cold and three heated), saving one tree every 1000 generations for a total of ten million generations. We excluded as ‘burn-in’ trees from the first two million generations, and tree distributions were checked for a stationary phase of likelihood. The posterior probabilities (PP) of clades were based on the majority-rule consensus produced with the remaining trees in MrBayes 3.1.2 [74].

### 4.4. Calibration

Estimates were conducted based on a simplified ultrametric Bayesian combined tree generated with BEAST 1.8.4 [75]. This analysis used a relaxed uncorrelated lognormal clock and Yule process speciation prior to inferring trees. The calibration parameters were based on previous estimates derived from a comprehensive fossil-calibrated study of the whole Malpighiaceae [44,76]. We opted for calibrating at the root, using a normal prior with mean initial values of 40.0 Mya (representing the age estimated for the MRCA of the Stigmaphylloid clade) and a standard deviation of 1.0 [44,67,76]. Two separate and convergent runs were conducted, with 10,000,000 generations, sampling every 1000 steps, and 2000 trees as burn-in. We checked for ESS values higher than 400 for all parameters on Tracer 1.6 [77]. Tree topology was assessed using TreeAnnotator and FigTree 1.4.0 [78].

### 4.5. Ancestral Area Reconstruction

Species distribution data were compiled from the taxonomic revision of *Bronwenia* [79], *Diplopterys* [80], and *Stigmaphyllon* [10,11] (Figure 3). Occurrences were categorized according to a modified version of the biome domains adopted by IBGE [81] and WWF [82], which reflects distribution patterns in *Stigmaphyllon*, namely: (A) Brazilian Atlantic Rainforest, (B) Seasonally Dry Tropical Forest [Caatinga], (C) Amazon Rainforest, (D) Cerrado, and (E) Australasian Rainforests (Figure 2). Ancestral areas of *Stigmaphyllon* and its relatives were estimated using a maximum likelihood analysis of geographic range evolution using software Rasp 3.2 [83]. Estimates were conducted using the statistical dispersal-extinction-cladogenesis (S-DEC) [84], according to the parameters proposed by Ree and Sanmartín [85].

### 4.6. Meta-Analysis

Ages of BAF lineages of vascular plants (ferns/lycophytes, gymnosperms, magnoliids, monocots, and eudicots) were compiled based on the phylogenetic literature and distribution data from Flora do Brasil [12] and Plants of the World Online [17]. Online repositories such as GBIF, BIEN and Species Link were not used since specimen identification is not usually updated or are not identified by a Malpighiaceae specialist. Additionally, the main problem in using all the above-mentioned repositories is that only Flora do Brasil present reliable information on the biome distribution of species sampled in our study. We considered a genus or lineage within a genus diversified in the BAF when at least 50% of its total number of species occurred in this biome. Data from BAF lineages of vascular plant species are presented in Table 2, alongside estimated ages, and references. A boxplot graphic is also presented in Figure 4, showing the diversification of vascular plants through time in the BAF based on data presented in Table 2.

## 5. Conclusions

Even though dispersal events from forested to open habitats have been recently identified as one of the main factors explaining richness in Neotropical biodiversity [16], the same pattern was not recovered for *Stigmaphyllon*. A late-Eocene origin for this genus is suggested, with its MRCA originating in the Northeastern BAFs, with several dispersal events taking place to other Neo- and Paleotropical biomes from 22.0 to 1.0 Mya, alongside several dispersals from Northern to Southern portions of the BAFs. When comparing our results with published divergence times for BAFs’ vascular plant lineages, a late-Eocene origin for these forests was evidenced. The immense gap in time-calibrated phylogenies focusing on BAFs’ vascular plant lineages is still the most significant impediment for a more comprehensive understanding of the plant diversification timeframe in these forests. Additionally, the recent evidenced tertiary uplift of Serra do Mar and Serra da Mantiqueira Mountain Ranges might also have played an important role in the diversification of the megadiverse BAFs.

## Figures and Tables

**Figure 1 plants-09-01569-f001:**
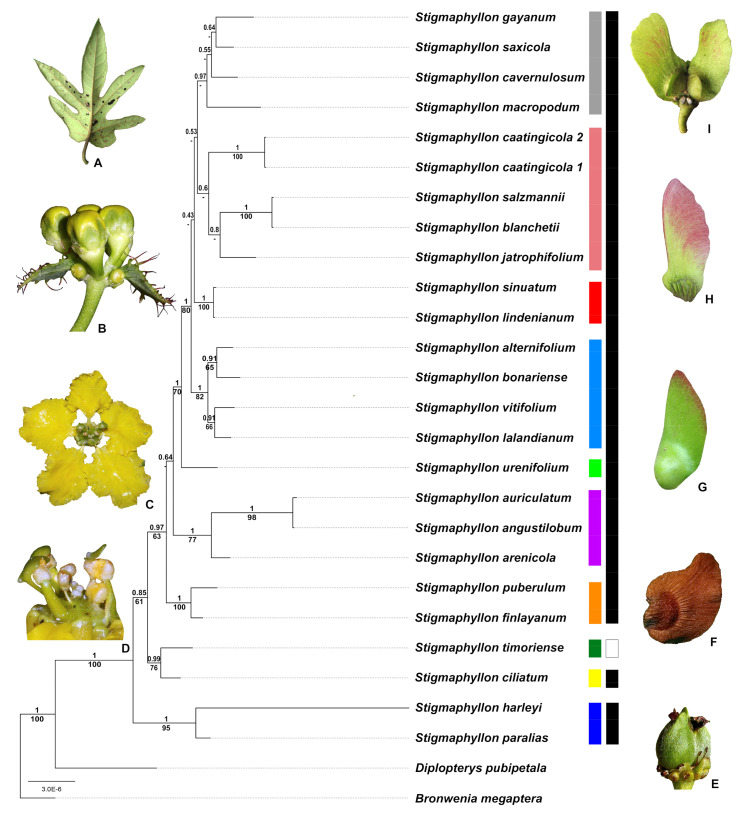
Phylogram from the combined analysis of nuclear and ribosomal markers for *Stigmaphyllon*. Posterior probabilities are shown above branches and bootstrap values below branches. Black bars represent *S.* subg. *Stigmaphyllon* and the white bar represents *S.* subg. *Ryssopterys*. (dark blue bar) *S. paralias* group; (yellow bar) *S. ciliatum* group; (dark green bar) *S.* subg. *Ryssopterys*; (orange bar) *S. finlayanum* group; (purple bar) *S. auriculatum* group; (light green bar) *S. urenifolium* group; (light blue bar) *S. lalandianum* group; (red bar) *S. sinuatum* group; (pink bar) *S. blanchetii* group; (grey bar) *S. saxicola* group. (**A**) adaxial surface of a leaf of *S. angustilobum* A.Juss. (**B**) Umbel of *S. ciliatum* (Lam.) A.Juss. in side view evidencing ciliate reduced leaves associated with the inflorescence. (**C**) open flower of *S. ciliatum* in frontal view. (**D**) androecium and gynoecium of *S. ciliatum* in side view, (**E**) mericarp of *S. paralias* A.Juss. in side view. (**F**) winged mericarp of *S. ciliatum* in side view. (**G**) winged mericarp of *S. auriculatum* (Cav.) A.Juss. in side view. (**H**) winged mericarp of *S. saxicola* C.E.Anderson in side view. (**I**) winged mericarps of *S. blanchetii* in side view (photographs by R.F.Almeida).

**Figure 2 plants-09-01569-f002:**
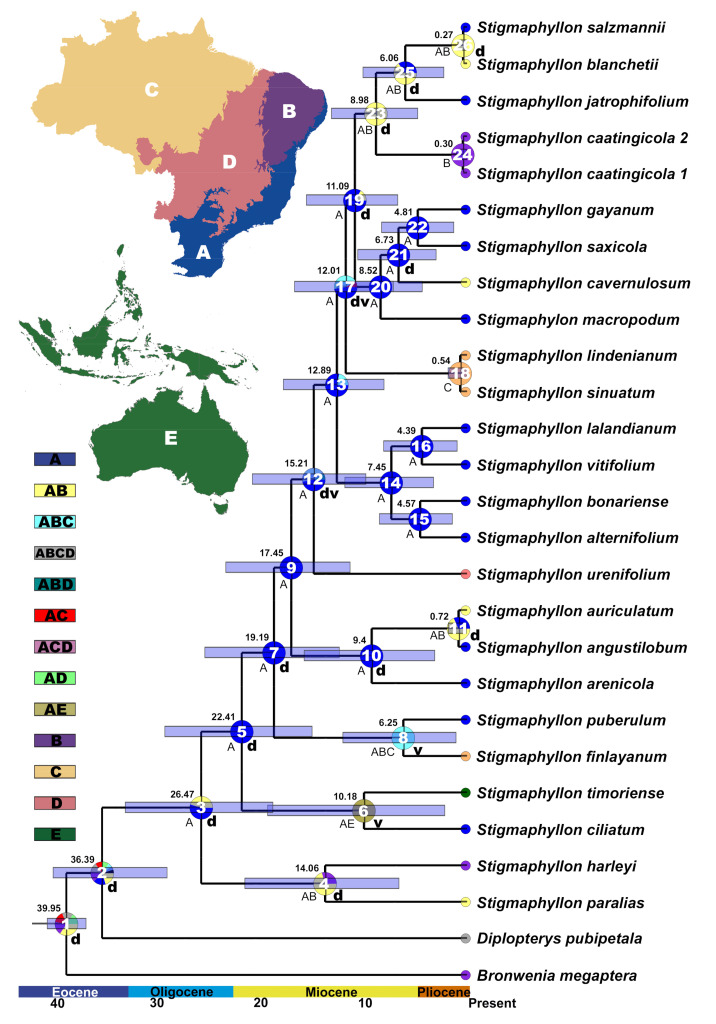
Chronogram and Statistical-Dispersal-Extinction-Cladogenesis (S-DEC) ancestral area reconstructions for *Stigmaphyllon*. Nodes are numbered from 1 to 26. Numbers above branches represent estimated ages (Mya). Branch letters on the left represent the reconstructed ancestral area(s): (**A**) Atlantic rainforest; (**B**) Seasonally Dry Tropical Forest; (**C**) Amazon rainforest; (**D**) Cerrado; (**E**) Australasian rainforests. Branch letters on the right represent dispersal (d) or vicariant (v) events.

**Figure 3 plants-09-01569-f003:**
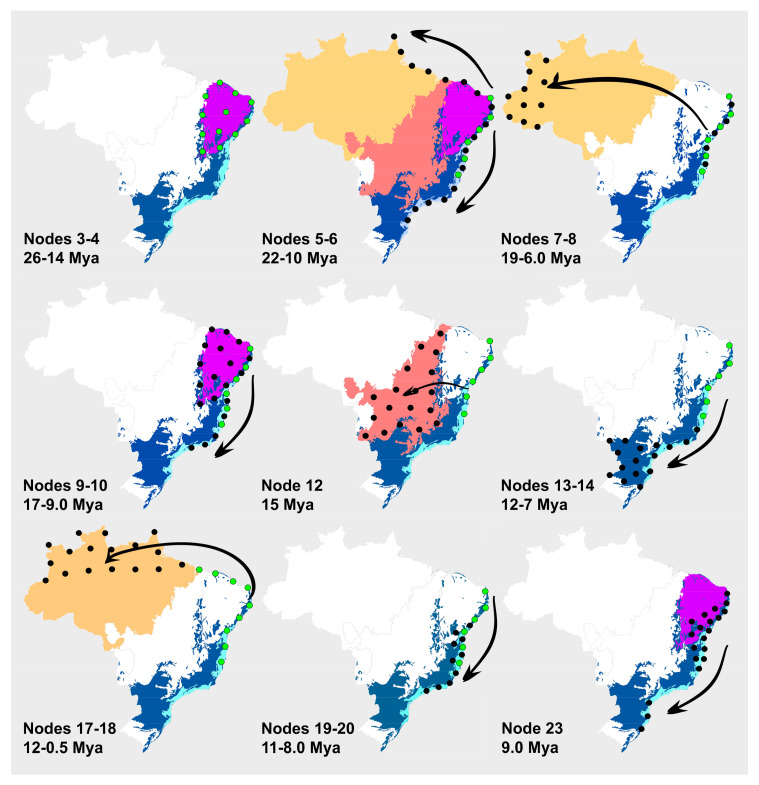
Times of dispersal/vicariance events leading to biome shifting in *Stigmaphyllon*. (blue) Atlantic rainforest; (purple) Seasonally Dry Tropical Forest; (orange) Amazon Rainforest; (red) Cerrado. Green circles represent the original ancestral populations of *Stigmaphyllon*. Dark circles represent populations of *Stigmaphyllon* colonizing new biomes over different time periods.

**Figure 4 plants-09-01569-f004:**
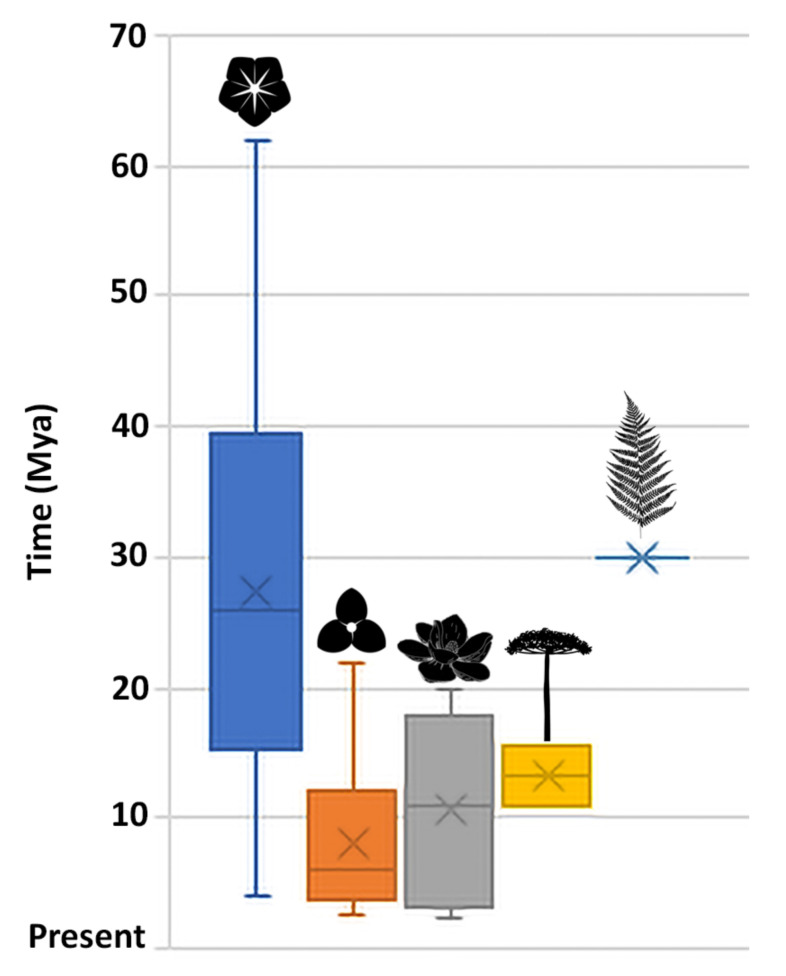
Box plots summarizing the diversification of vascular plants through time in the Brazilian Atlantic Rainforest based on data presented in Table 2. (blue) Eudicots; (orange) Monocots; (gray)–Magnoliids; (yellow) Gymnosperms; (light blue) Ferns.

**Table 1 plants-09-01569-t001:** Divergence times estimates (maximum/mean/minimum), ancestral area reconstructions, and dispersal/vicariance events reconstructed for all nodes in this study.

Nodes	Max	Age Mean	Min	Ancestral Area Reconstruction	Dispersal/Vicariance Event
1	40.0	39.95	35.0	Atlantic Forest/Caatinga	Dispersal
2	41.29	36.39	29.92	Atlantic Forest/Caatinga	Dispersal
3	34.09	26.47	19.35	Atlantic Forest	Dispersal
4	22.14	14.06	6.76	Caatinga	Dispersal
5	30.14	22.41	15.43	Atlantic Forest	Dispersal
6	19.87	10.18	2.18	Atlantic Forest/Asian Rainforests	Vicariance
7	26.14	19.19	12.71	Atlantic Forest	Dispersal
8	12.35	6.25	1.06	Atlantic Forest/Caatinga/Amazon Forest	Vicariance
9	24.05	17.45	11.64	Atlantic Forest	-
10	16.22	9.4	3.18	Atlantic Forest	Dispersal
11	1.81	0.72	0.03	Atlantic Forest/Caatinga	Dispersal
12	21.40	15.21	10.06	Atlantic Forest	Dispersal/Vicariance
13	18.30	12.89	8.25	Atlantic Forest	-
14	12.16	7.45	3.30	Atlantic Forest	-
15	8.70	4.57	1.42	Atlantic Forest	-
16	8.29	4.39	0.95	Atlantic Forest	-
17	17.19	12.01	7.32	Atlantic Forest	Dispersal/Vicariance
18	1.84	0.54	0.0	Amazon Forest	-
19	16.0	11.09	6.88	Atlantic Forest	Dispersal
20	13.05	8.52	4.43	Atlantic Forest	-
21	10.87	6.73	3.04	Atlantic Forest	Dispersal
22	8.49	4.81	1.28	Atlantic Forest	-
23	13.49	8.98	4.88	Atlantic Forest/Caatinga	Dispersal
24	0.36	0.30	0.10	Caatinga	-
25	10.35	6.06	2.27	Atlantic Forest/Caatinga	Dispersal
26	0.86	0.27	0.0	Atlantic Forest/Caatinga	Dispersal

**Table 2 plants-09-01569-t002:** Age of Brazilian Atlantic Rainforests (BAF) lineages of vascular plants based on phylogenetic literature and distribution data from Flora do Brasil [13] and Plants of the World Online [17]. ? refers to ages unavailable from the consulted literature.

Family	Genus/Lineage	BAF spp./Genus	Phytophysiognomy	Max	Mean	Min	Reference
**Ferns**							
Cyatheaceae	*Cyathea* Sm.	23/290	Rainforest	?	30.0	?	[18]
**Gymnosperms**							
Araucariaceae	*Araucaria* Juss.	1/20	Rainforest	?	10.0	?	[19]
Podocarpaceae	*Podocarpus* L’Hér. ex Pers.	2/115	Rainforest	?	15.0	?	[20]
**Magnoliids**							
Annonaceae	*Hornschuchia* Nees	10/10	Rainforest	?	20.0	?	[21]
Lauraceae	*Phyllostemonodaphne* Koesterm.	1/1	Rainforest	?	?	?	-
Lauraceae	*Urbanodendron* Mez	3/3	Rainforest	?	?	?	-
Monimiaceae	*Macropeplus* Perkins	4/4	Rainforest	?	11.26	?	[22]
Monimiaceae	*Grazielanthus* Peixoto and Per.-Moura	1/1	Rainforest	?	3.85	?	[22]
Monimiaceae	*Hennecartia* J.Poiss.	1/1	Rainforest	?	15.58	?	[22]
Monimiaceae	*Macrotorus* Perkins	1/1	Rainforest	?	11.26	?	[22]
Monimiaceae	*Mollinedia* Ruiz and Pav.	32/55	Rainforest	?	2.20	?	[22]
**Monocots**							
Amaryllidaceae	*Griffinia* Ker Gawl.	17/22	Grassland	?	?	?	-
Araceae	*Asterostigma* Fisch. and C.A. Mey.	8/10	Rainforest	?	?	?	-
Araceae	*Dracontioides* Engl.	2/2	Rainforest	?	?	?	-
Asparagaceae	*Herreria* Ruiz and Pav.	6/8	Rainforest	?	?	?	-
Bromeliaceae	*Alcantarea* (E.Morren ex Mez) Harms	30/40	Rainforest	5.5	3.3	1.5	[23]
Bromeliaceae	*Araeococcus* Brongn.	6/9	Rainforest	?	3.5	?	[24]
Bromeliaceae	*Billbergia* Thunb.	35/63	Rainforest	?	4.5	?	[24]
Bromeliaceae	*Canistropsis* (Mez) Leme	11/12	Rainforest	?	3.5	?	[24]
Bromeliaceae	*Stigmatodon* Leme, G.K.Br. and Barfuss	18/18	Rainforest	6.4	5.5	2.8	[23]
Bromeliaceae	*Vriesea* Lindl.	167/255	Rainforest	6.8	5.0	3.3	[23]
Commelinaceae	*Siderasis* Raf.	6/6	Rainforest	16.69	8.57	2.26	[25]
Commelinaceae	*Dichorisandra* J.C.Mikan	40/52	Rainforest	6.38	2.78	0.32	[25]
Dioscoreaceae	*Dioscorea* L.	81/628	Rainforest	30.0	22.0	15.0	[26]
Iridaceae	*Neomarica* Sprague	27/27	Grassland	?	6.5	?	[27]
Marantaceae	*Ctenanthe* Eichler	11/15	Rainforest	?	?	?	-
Marantaceae	*Maranta* L.	20/37	Rainforest	?	?	?	-
Marantaceae	*Saranthe* Eichler	8/10	Rainforest	?	?	?	-
Marantaceae	*Thalia* L.	4/6	Rainforest	?	?	?	-
Orchidaceae	*Bifrenaria* Lindl.	17/21	Rainforest	?	13.0	?	[28]
Orchidaceae	*Capanemia* Barb.Rodr.	6/9	Rainforest	?	?	?	-
Orchidaceae	*Centroglossa* Barb.Rodr.	6/6	Rainforest	?	?	?	-
Orchidaceae	*Cirrhaea* Lindl.	7/7	Rainforest	?	?	?	-
Orchidaceae	*Hoehneella* Ruschi	2/2	Rainforest	?	?	?	-
Orchidaceae	*Isabelia* Barb. Rodr.	3/3	Rainforest	?	?	?	-
Orchidaceae	*Lankesterella* Ames	7/11	Rainforest	?	?	?	-
Orchidaceae	*Loefgrenianthus* Hoehne	1/1	Rainforest	?	?	?	-
Orchidaceae	*Miltonia* Lindl.	19/19	Rainforest	?	?	?	-
Orchidaceae	*Phymatidium* Lindl.	9/9	Rainforest	?	?	?	-
Orchidaceae	*Pseudolaelia* Porto and Brade	10/15	Rainforest	?	?	?	-
Poaceae	*Chusquea* Kunth	45/185	Rainforest	?	9.0	?	[29]
Poaceae	*Merostachys* Spreng.	44/53	Rainforest	?	?	?	-
Poaceae	*Olyra* L.	9/15	Rainforest	?	14.0	?	[30]
Poaceae	*Raddia* Bertol.	9/12	Rainforest	?	22.0	?	[30]
**Eudicots**							
Acanthaceae	*Herpetacanthus* Nees	14/21	Rainforest	?	?	?	-
Apocynaceae	*Bahiella* J.F.Morales	2/2	Rainforest	?	?	?	-
Apocynaceae	*Peplonia* Decne.	8/13	Rainforest	?	13.0	?	[31]
Asteraceae	*Barrosoa* R.M.King and H.Rob.	7/11	Rainforest	?	?	?	-
Asteraceae	*Disynaphia* Hook. and Arn. ex DC.	10/14	Rainforest	?	?	?	-
Asteraceae	*Grazielia* R.M.King and H.Rob.	7/12	Rainforest	?	?	?	-
Asteraceae	*Pamphalea* DC.	8/9	Rainforest	?	?	?	-
Asteraceae	*Stifftia* J.C.Mikan	4/6	Rainforest	?	34.0	?	[32]
Asteraceae	*Piptocarpha* R.Br.	24/50	Rainforest	?	?	?	-
Bignoniaceae	*Dolichandra* Cham.	8/9	Rainforest	38.3	31.3	25.2	[33]
Bignoniaceae	*Paratecoma* Kuhlm.	1/1	Rainforest	?	?	?	-
Bignoniaceae	*Zeyheria* Mart.	2/2	Rainforest	?	?	?	-
Cactaceae	*Rhipsalis* Gaertn.	37/43	Rainforest	11.82	7.67	4.26	[34]
Callophyllaceae	*Kielmeyera* Mart. and Zucc.	24/50	Rainforest	23.0	15.54	10.0	[35]
Cleomaceae	*Tarenaya* Raf.	14/14	Rainforest	18.39	16.60	14.7	[36]
Clusiaceae	*Tovomitopsis* Planch. and Triana	2/2	Rainforest	34.0	20.46	13.0	[37]
Erythroxylaceae	*Erythroxylum* P.Browne	71/259	Rainforest	?	?	?	-
Euphorbiaceae	*Brasiliocroton* P.E.Berry and Cordeiro	2/2	Rainforest	58.42	48.13	39.6	[38]
Euphorbiaceae	*Astraea* Klotzsch	10/14	Rainforest	34.38	19.05	4.46	[38]
Euphorbiaceae	*Croton* L.	98/1157	Rainforest	40.92	39.03	37.0	[38]
Fabaceae	*Dahlstedtia* Malme	9/16	Rainforest	?	?	?	-
Fabaceae	*Holocalyx* Micheli	1/1	Rainforest	52.1	1.3	28.8	[39]
Fabaceae	*Moldenhawera* Schrad.	8/11	Rainforest	?	48.0	?	[40]
Fabaceae	*Parapiptadenia* Brenan	5/6	Rainforest	?	11.0	?	[41]
Fabaceae	*Paubrasilia* Gagnon, H.C.Lima and G.P.Lewis	1/1	Rainforest	?	48.0	?	[41]
Fabaceae	*Schizolobium* Vogel	1/1	Rainforest	?	40.5	?	[41]
Gentianaceae	*Calolisianthus* (Griseb.) Gilg	4/4	Rainforest	?	?	?	-
Gentianaceae	*Chelonanthus* (Griseb.) Gilg	4/5	Rainforest	?	?	?	-
Gentianaceae	*Deianira* Cham. and Schltdl.	4/7	Rainforest	?	?	?	-
Gentianaceae	*Prepusa* Mart.	5/6	Rainforest	?	?	?	-
Gentianaceae	*Senaea* Taub.	1/2	Rainforest	?	?	?	-
Gentianaceae	*Tetrapollinia* Maguire and B.M.Boom	1/1	Rainforest	?	?	?	-
Gesneriaceae	*Codonanthe* (Mart.) Hanst.	8/9	Rainforest	?	7.0	?	[42]
Gesneriaceae	*Nematanthus* Schrad.	32/32	Rainforest	?	7.0	?	[42]
Gesneriaceae	*Paliavana* Vand.	4/6	Rainforest	?	7.0	?	[42]
Gesneriaceae	*Sinningia* Nees	70/75	Rainforest	?	14.0	?	[42]
Gesneriaceae	*Vanhouttea* Lem.	9/10	Rainforest	?	7.0	?	[42]
Loasaceae	*Blumenbachia* Schrad.	7/12	Rainforest	4.22	27.5	20.7	[43]
Malpighiaceae	*Barnebya* (Griseb.) W.R.Anderson and B.Gates	1/2	Rainforest	?	62.0	?	[44]
Melastomataceae	*Behuria* Cham.	17/17	Rainforest	?	8.50	?	[45]
Melastomataceae	*Bertolonia* Raddi	27/27	Rainforest	?	9.26	?	[45]
Melastomataceae	*Huberia* DC.	13/17	Rainforest	?	8.50	?	[45]
Melastomataceae	*Physeterostemon* R.Goldenb. and Amorim	5/5	Rainforest	?	2.50	?	[45]
Melastomataceae	*Pleiochiton* Naudin ex A. Gray	12/12	Rainforest	?	4.43	?	[45]
Melastomataceae	*Pleroma* D.Don	46/85	Rainforest	?	12.26	?	[45]
Moraceae	*Clarisia* Ruiz and Pav.	2/2	Rainforest	?	?	?	-
Moraceae	*Ficus* L.	38/874	Rainforest	?	25.0	?	[46]
Myrtaceae	*Accara* Landrum	1/1	Rainforest	?	32.0	?	[47]
Myrtaceae	*Blepharocalyx* O.Berg.	3/4	Rainforest	?	40.0	?	[47]
Myrtaceae	*Calyptranthes* Sw.	35/35	Rainforest	?	24.0	?	[47]
Myrtaceae	*Curitiba* Salywon and Landrum	1/1	Rainforest	?	40.0	?	[47]
Myrtaceae	*Eugenia* L.	254/1149	Rainforest	?	44.0	?	[47]
Myrtaceae	*Myrceugenia* O.Berg.	33/45	Rainforest	?	25.0	?	[47]
Myrtaceae	*Myrcia* DC.	200/609	Rainforest	?	28.0	?	[48]
Myrtaceae	*Myrciaria* O.Berg.	18/27	Rainforest	?	26.0	?	[48]
Myrtaceae	*Myrrhinium* Schott	1/1	Rainforest	?	29.0	?	[47]
Myrtaceae	*Neomitranthes* D.Legrand	15/15	Rainforest	?	12.0	?	[47]
Myrtaceae	*Plinia* L.	29/78	Rainforest	?	37.0	?	[47]
Myrtaceae	*Psidium* L.	39/91	Rainforest	?	20.0	?	[47]
Rubiaceae	*Bathysa* C. Presl	6/10	Rainforest	?	?	?	-
Rubiaceae	*Bradea* Standl. ex Brade	6/6	Rainforest	?	?	?	-
Rubiaceae	*Coccocypselum* P.Browne	14/22	Rainforest	?	?	?	-
Rutaceae	*Conchocarpus* J.C.Mikan	40/48	Rainforest	?	?	?	-
Rutaceae	*Metrodorea* A.St.-Hil	3/6	Rainforest	?	?	?	-
Rutaceae	*Neoraputia* Emmerich ex Kallunki	5/6	Rainforest	?	?	?	-
Sapindaceae	*Cardiospermum* L.	6/10	Rainforest	?	23.0	?	[49]
Sapindaceae	*Thinouia* Triana and Planch.	7/11	Rainforest	?	?	?	-
Solanaceae	*Petunia* Juss.	10/16	Grassland	11.5	8.49	5.5	[50]
Vochysiaceae	*Callisthene* Mart.	8/8	Rainforest	30.0	22.0	14.0	[51]

**Table 3 plants-09-01569-t003:** Species and DNA regions sampled in this study. Genbank accession numbers are presented for columns ETS (External Transcribed Spacer), ITS (Internal Transcribed Spacer), and PHYC (Phytochrome C gene). * Sequences were obtained from GenBank.

Species	Voucher (Herbarium Acronym)	ETS	ITS	PHYC
*Bronwenia cinerascens* (Benth.) W.R.Anderson and C.C.Davis	*Nee* 48, 658 (SP)	KR054586.1	HQ246821.1 *	HQ246987.1 *
*Diplopterys pubipetala* (A.Juss.) W.R.Anderson and C.C.Davis	*Francener* 1126 (SP)	KR092986	HQ246821.1 *	HQ247045.1 *
*Stigmaphyllon alternifolium* A.Juss.	*Almeida* 501 (SP)	KR054612.1	-	-
*Stigmaphyllon angustilobum* A.Juss.	*Almeida* 503 (SP)	KR054591.1	MT559811	-
*Stigmaphyllon arenicola* C.E.Anderson	*Sebastiani* 5 (SP)	KR054589.1	-	-
*Stigmaphyllon auriculatum* (Cav.) A.Juss.	*Almeida* 584 (HUEFS)	KR054592.1	MT559812	-
*Stigmaphyllon blanchetii* C.E.Anderson	*Almeida* 525 (SP)	KR054615.1	MT559813	-
*Stigmaphyllon bonariense* A.Juss.	*Queiroz* 13, 530 (HUEFS)	KR054607.1	-	-
*Stigmaphyllon caatingicola* 1 R.F.Almeida and Amorim	*Almeida* 577 (HUEFS)	KR054595.1	MT559814	-
*Stigmaphyllon caatingicola* 2 R.F.Almeida and Amorim	*Almeida* 629 (HUEFS)	MT490608	MT559815	-
*Stigmaphyllon cavernulosum* C.E.Anderson	*Cardoso* 2083 (HUEFS)	KR054609.1	-	-
*Stigmaphyllon ciliatum* (Lam.) A.Juss.	*Almeida* 541 (SP)	KR054590.1	-	HQ247151.1 *
*Stigmaphyllon finlayanum* A.Juss.	*Pace* 457 (SPF)	KR054599.1	-	HQ247152.1 *
*Stigmaphyllon gayanum* A.Juss.	*Almeida* 500 (SP)	KR054610.1	-	-
*Stigmaphyllon harleyi* C.E.Anderson	*Santos* 378 (HUEFS)	KR054594.1	MT581513	-
*Stigmaphyllon jatrophifolium* A.Juss.	*Filho s.n.* (SP369102)	KR054614.1	-	-
*Stigmaphyllon lalandianum* A.Juss.	*Kollmann* 4279 (CEPEC)	KR054596.1	-	-
*Stigmaphyllon lindenianum* A.Juss.	*Aguillar* 718 (SP)	KR054603.1	-	HQ247153.1 *
*Stigmaphyllon macropodum* A.Juss.	*Almeida* 538 (SP)	KR054602.1	MT559816	-
*Stigmaphyllon paralias* A.Juss.	*Almeida* 509 (SP)	KR054593.1	KY421909.1	AF500566.1 *
*Stigmaphyllon puberulum* A.Juss.	*Perdiz* 732 (HUEFS)	KR054600.1	-	-
*Stigmaphyllon salzmannii* A.Juss.	*Almeida* 526 (SP)	KR054616.1	MT559817	-
*Stigmaphyllon saxicola* C.E.Anderson	*Badini* 24,261 (HUEFS)	KR054605.1	MT559818	-
*Stigmaphyllon sinuatum* (DC.) A.Juss.	*Amorim 3159* (CEPEC)	KR054608.1	-	-
*Stigmaphyllon timoriense* (DC) C.E.Anderson	*Anderson* 796 (US)	-	-	AF500545.1 *
*Stigmaphyllon urenifolium* A.Juss.	*Guedes* 13,932 (HUEFS)	KR054604.1	-	-
*Stigmaphyllon vitifolium* A.Juss.	*DalCol* 233 (HUEFS)	KR054598.1	-	-

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
