# Peer review of "Biogeography of Stigmaphyllon (Malpighiaceae) and a Meta-Analysis of Vascular Plant Lineages Diversified in the Brazilian Atlantic Rainforests Point to the Late Eocene Origins of This Megadiverse Biome"

_plants, 2020, doi:10.3390/plants9111569_

Round 1

Reviewer 1 Report

The manuscript presents a time-calibrated phylogenetic classification of the genus Stigmaphyllon from the Brazilian Atlantic Rainforest. It is an interesting topic that would support many aspects of future studies in Stigamaphyllon. In the Abstract, the authors should briefly mention the need for the research and how this study could provide a baseline for future research. The introduction is too short and requires strict revision with particulars. Other sections are well-composed. Due to the PDF version of the file, I'm not able to correct the language and grammar that exist. 

Author Response

Response to Reviewer 1 Comments

Point 1: In the Abstract, the authors should briefly mention the need for the research and how this study could provide a baseline for future research.

Response 1: We added a phrase in the abstract stating the need for the research and how our study could provide a baseline for future research.

Point 2: The introduction is too short and requires strict revision with particulars.

Response 2: We re-wrote the introduction to make it more inclusive and better justify our analyses, including the meta-analysis.

Point 3: Other sections are well-composed.

Response 3: Thank you.

Point 4: Due to the PDF version of the file, I'm not able to correct the language and grammar that exist.

Response 4: Thank you for noticing. We used a grammar corrector to find all language and grammar typos in the text.

Reviewer 2 Report

Dear authors,

You present an interesting study regarding the biogeography and phylogeny of Stigmatophyllon. I've made several comments in the attached pdf, which aim at improving your manuscript. I hope that you will find them useful. 

Author Response

Response to Reviewer 2 Comments

Page 1

Point 1: In the Materials and Methods section you state that you sampled 24 species.

Response 1: We included an explanation of our sampling in the Material and Methods section. A total of 24 species of Stigmaphyllon were sampled: 18 spp. from the Atlantic Rainforest, 3 spp. from the Amazon Rainforest, 2 spp. com the Caatinga domain, and 1 spp. from the Cerrado domain.

Point 2: Mya I guess.

Response 2: We agree. The abbreviation Mya was added by the end of the sentence.

Point 3: Ditto

Response 3: We agree. The abbreviation Mya was added by the end of the sentence.

Point 4: 'and a' instead of ';'

Response 4: We agree and replaced ; “by and a”.

Point 5: what do you mean by that?

Response 5: We mean vascular plant lineages that show clades restricted to the Brazilian Atlantic Rainforest. To ensure a better understanding, we changed the wording in this sentence to “64 lineages of vascular plants diversified in the Brazilian Atlantic Rainforest”. 

Point 6: This seems like a bold statement. Maybe consider rephrasing it

Response 6: We kindly disagree. No study to date has ever focused on the possible Tertiary origins of the Brazilian Atlantic Rainforests from the plant chronograms point of view. Thus, our results are the first empirical ones to address such topic.

Point 7: Maybe you mean diverse?

Response 7: Yes, we mean diverse, instead of diversified. We changed the wording to diverse in this sentence.

Page 2

Point 8: Not confined in the tropics and the subtropics then? Maybe consider rephrasing this sentence, to be more clear. You could say that you focus on this genus, because you want to provide insights regarding its biogeography, which has been overlooked, in contrast to its taxonomy. Of course, your manuscript would benefit if you could provide more details regarding the biogeographical works that might have been conducted in the wider region. It wouldn't hurt either if you introduced the reader more to the BAF biogeography and provided more info for BAF in general (why it's important, what's so special about it, etc). The Introduction section seems rather short and abrupt. You could take inspiration from the following paper regarding the BAF details: Leão, Tarciso CC, Eimear Nic Lughadha, and Peter B. Reich. "Evolutionary patterns in the geographic range size of Atlantic Forest plants." Ecography (2020).

Response 8: We agree. In order to make our goals clearer to the reader we rewrote the last paragraph: “Previous phylogenetic studies for Malpighiaceae suggested a Brazilian Atlantic Rainforest (herein treated as BAF) origin for Stigmaphyllon, with S. paralias A.Juss. being consistently recovered in several studies as the first lineage to diverge in the genus [6,2]. Even though the BAFs currently comprises more than 15.000 known vascular plant species [75], they are regarded as one of the most threatened hotspots for biodiversity worldwide due to being mostly fragmented and disturbed by most of Brazil’s human population and economic activity [77]. In this study, we focus on the biogeography history of the BAFS using Stigmaphyllon as a model, allied to a meta-analysis of vascular plant lineages diversified in this biome to hypothesize the age of BAFs. More specifically, we: (1) test the monophyly of Stigmaphyllon and its subgenera; (2) time-calibrate the phylogenetic tree; (3) estimate the ancestral areas of Stigmaphyllon; and (4) shed some light on the age and biogeographic history of the BAFs using Stigmaphyllon and a meta-analysis of vascular plant lineages diversified in this biome.”

Point 9: You do not state anywhere in the Introduction that you are going to conduct a meta-analysis (and to be fair, it does not seem like you conducted a meta-analysis either) or why you wanted to conduct a meta-analysis.

Response 9: We agree and rewrote the last paragraph of the introduction to better explain to the reader why we performed a meta-analysis. Please, see response 8.

Point 10: What do these abbreviations stand for? You present them in the Materials & Methods section, which comes in later.

Response 10: We rephrased this sentence to “Our combined dataset for ribosomal (ETS, ITS) and nuclear (PHYC) markers contains a total of 2283 characters.”

Point 11: What do you mean by that? (I am referring to the 'combined' phrasing).

Response 11: We mean the ribosomal + nuclear datasets, and rephrased it to “The combined analysis of nuclear and ribosomal markers provides…”

Point 12: What do these abbreviations stand for?

Response 12: Full names, Maximum Parsimony and Bayesian Inference, were presented before these abbreviations to make it clearer to the reader.

Point 13: replace ( with [.

Response 13: Agreed. We replaced ( with [.

Point 14: replace ) with ].

Response 14: We just added ] after ).

Page 3

Point 15: It would be nice if you informed the readers which plant part is depicted in each sub-figure.

Response 15: Agreed. We added descriptions for each morphological detail presented in this figure.

Page 4

Point 16: What does MRCA stand for?

Response 16: It stands for most recent common ancestor. We added its full meaning before the abbreviation in the text.

Point 17: What is SDTF?

Response 17: It means Seasonally Dry Tropical Forests, and its full meaning was added before the abbreviation in the text.

Point 18: This is the Chibanian stage of the Pleistocene and definitely not the Holocene, which started 11.7 Kya.

Response 18: We agree and replaced Holocene by Pleistocene in the text.

Point 19: Delete Age from these three columns and place it above them in one cell

Response 19: Agreed. We made the abovementioned changes in the table.

Page 6

Point 20: See my other comment for this figure.

Response 20: Please, see responses 21-24.

Point 21: These are not readable, please consider choosing a different colour palette

Response 21: We kindly disagree. The colour palette is quite distinguishable in the tree pie charts. The main problem is that, sometimes, percentages for some areas are not significant enough to be clearly represented in the pie charts. Thus, in order to make the visualization of ancestral areas reconstruction clearer for readers, we added letters on the left representing the code for the most likely ancestral are constructed.

Point 22: So this means a dispersal event?

Response 22: Yes, after the abovementioned changes were made in this figure, d/v on the right of a node means a dispersal (d) or vicariant (v) event.

Point 23: And this refers to the Cerrado biome?

Response 23: No, this refers to the Amazon Rainforest.

Point 24: Consider changing which letters are upper and which are lower, since there is some confusion with your figure (see my other comments).

Response 24: Agreed. After the abovementioned changes in this figure, upper letters represent ancestral areas, and lower letters represent dispersal or vicariant events.

Page 7

Point 25: What do the (nearly impossible to see) white dots represent?

Response 25: White dots in this figure represent ancestral Atlantic Rainforest populations of Stigmaphyllon lineages. An explanation was added to this figure’s caption.

Point 26: Do the black dots represent your occurrences? They seem gridded. If so, you need to take some measures against any possible spatial aggregation issues.

Response 26: No, black dots represent populations of Stigmaphyllon colonizing biomes or new parts of a biome through time. An explanation was added to this figure’s caption.

Point 27: How did you account for the variability in the estimation of phylogenetic divergence time for each species in your list?

Response 27: Most studies used the software Beast to estimate divergence times for these lineages. The graphical representation of these estimates on a chronogram are represented by standard deviation bars. Thus, we recorded maximum and minimum ages represented by these bars, when the chronogram was presented in a good graphical quality that allowed this visualization. Additionally, few studies present a table with all maximum, mean, and minimum divergence times values estimated for each node of the chronogram. 

Point 28: Are these species or species complexes? How many species in total are they?

Response 28: None. Lineages here represent genera or clades within a genus that are mostly composed by Atlantic Rainforest species. 

Page 12

Point 29: What do you mean by that?

Response 29: We rephrased here as “The topology recovered from the combined dataset (i.e., ribosomal + nuclear markers).”

Page 14

Point 30: In the Abstract you say that you used 18 species. Please clarify.

Response 30: We rephrased here as “We sampled 24 species of Stigmaphyllon (ca. ¼ of the Neotropics' genus diversity: 18 spp. from the Brazilian Atlantic Rainforest, three spp. from the Amazon Rainforest, two spp. from the Caatinga, and one spp. from the Cerrado biomes).”

Point 31: Please state what the abbreviations mean in the table columns.

Response 31: Agreed. A description for these abbreviations was added in the table’s caption.

Page 16

Point 32: Why not from GBIF, BIEN and the Species Link Network as well? Also, why not using data from Flora do Brasil and PWO as you state in the next paragraph?

Response 32: In RFAlmeida’s experience as a Malpighiaceae taxonomist, specimen identifications in repositories such as GBIF, BIEN and Species Link are not updated or, sometimes, unreliable due to not being identified by a specialist in the group. RFAlmeida is responsible for elaborating the monograph of Brazilian Stigmaphyllon in Flora do Brasil, and, unfortunately due to the COVID pandemic, the monograph is not yet completed. Additionally, the main problem in using all the above-mentioned repositories is that only Flora do Brasil present reliable information on the biome distribution of species sampled in our study. 

Point 33: Could you please explain the reason behind choosing this model?

Response 33: Agreed. We rephrased this sentence to “Estimates were conducted using the statistical dispersal-extinction-cladogenesis (S-DEC) [74], according to the parameters proposed by Ree and Sanmartín [79].”

Point 34: Maybe consider changing the title here. It does not seem relevant.

Response 34: We kindly disagree. The results and discussion from the meta-analysis are one of the cores of our study.

Page 17

Point 35: Such as? Did you use the tree from Smith & Brown (2018)? Please provide here more details, as it is crucial for your analysis and results. Smith, Stephen A., and Joseph W. Brown. "Constructing a broadly inclusive seed plant phylogeny." American journal of botany 105, no. 3 (2018): 302-314.

Response 35: All literature used for the meta-analysis is presented on table 2. Unfortunately, we did not use the tree from the abovementioned study since the sampling focused on families and orders of flowering plants.

Point 36: I guess this section could go before the Materials and Methods, because reading the manuscript was really confusing and hard with its current structure.

Response 36: Agreed. We moved up this section and placed it after the discussion and before the Material and Methods section.

Reviewer 3 Report

see attachment

Author Response

Response to Reviewer 3 Comments

Point 1: The paper is mixing up two topics, the time calibration and biogeography of Stigmaphyllon and a meta-analysis of diverse vascular plant lineages. It is not clear why both topics are covered. It is not mentioned in the introduction and in the aims of the paper. For my opinion, the focus should be on Stigmaphyllon only, the other part should be removed.

Response 1: We kindly disagree. In a biogeographic study, the main focus is the distribution pattern, in our case the Atlantic Rainforest biome, and not the biological entity. Thus, our goal was to reconstruct part of the biogeographic history of the Atlantic Rainforests using Stigmaphyllon as a study model. However, since a comprehensive analysis reporting the estimated ages for vascular plant lineages diversified in the Atlantic Rainforest is completely absent from the literature, we felt the need to perform such analysis to properly discuss the age of the Atlantic Rainforest. Additionally, our results will be the baseline for future studies on the diversification of this biome.

Also, the inclusion of a broader topic such as the meta-analysis will upgrade much the value of the paper to a broader audience in Plants, and for this reason we decided to submit this paper which include both topics (a more limited data-based results of Stigmaphyllon as a baseline to include a comparison with other lineages). This makes the paper interesting to all researchers dealing with the biogeography of plants and animals in the Atlantic Forest and potentially increases much the value of the paper and its citations.

Point 2: There are many papers available in which the phylogeny of Stigmaphyllon is covered which are referenced. It should be clearly described what’s new. This is more or less written in the aims but not fully clear from the introduction. The focus should be on ancestral area reconstruction and divergence times estimation.

Response 2: We kindly disagree. As stated in our introduction, this is the first phylogenetic study focusing solely on Stigmaphyllon. We also state that our study is the first to properly test the monophyly of the genus, since all previous studies (all of them at the family level and sampling few species of Stigmaphyllon) did not sample the type specimen of Stigmaphyllon. Additionally, it is not possible to reconstruct the biogeographic history of a genus without testing its monophyly first. Otherwise, our results would be far from the probable truth.

Point 3: Figure 1: We have three subgenera in the figure but only two in the text, what about subg. Brachypteris? There are some morphological characters pictured but not described. Is there a support of the phylogram by the morphological characters? The quality of the images could be improved.

Response 3: The first version of this manuscript included character mapping of several morphological characters + the proposition of a new infrageneric classification for Stigmaphyllon. However, the manuscript, in this circumscription, was too long for publication. So, we opted for dealing with these morphological analyses elsewhere, but, unfortunately, forgot to erase some information from figure 1. We apologize for that and present a corrected version of figure 1. Additionally, the quality of the figure is low since we used its jpeg version in the text. If needed, we have tiff versions of all figures with 600 dpi of resolution.

Point 4: Figure 3: The quality of the figure should be improved. What is described, Brazil, South America (also figure 2)? The figure 3 is mentioned only under methods. It is not used in the results.

Response 4: We kindly disagree. The quality of the figure is low since we used its jpeg version in the text. If needed, we have tiff versions of all figures with 600 dpi of resolution. The political geography depicted in figure 3 is that of Brazil since our study focuses only on the biogeographical history of the Brazilian Atlantic Rainforest. Additionally, we cited figure 3 along the result section of the text. Thank you for noticing.

Point 5: Material/Methods: 4.1: How many samples per species are studied? For a detailed phylogeny one sample per species is not enough. 4.4: There are calibration parameters referenced. However, it would be good to describe the fossil records and the basis of the calibration.

Response 5: As presented on table 1, we sampled a single specimen for each species, except for Stigmaphyllon caatingicola, which we had a second sample. We agree that one sample per species is not enough for a phylogenetic study when species complexes, their taxonomy or species level problems are the main focus. However, our main focus was to reconstruct the historical biogeography at the generic/subclade level to understand where the genus originated, when and how many times it left its original biome, and how its lineages diversified through time in relation to neighbouring biomes in South America. Additionally, as mentioned in our material and methods section, all calibration parameters follow published calibration trees for a generic level phylogeny of Malpighiaceae. Thus, we have used calibration points derived from these abovementioned studies in our study. Currently, there are no record of fossils for Stigmaphyllon.

Round 2

Reviewer 2 Report

Dear authors,

You present an interesting study regarding the biogeography and phylogeny of Stigmatophyllon. Thank you for addressing most of the issues raised. I've made some comments in the attached pdf, which aim at further improving your manuscript (please consider expanding the Introduction section and providing more context regarding the need of conducting a meta-analysis). I hope that you will find them useful. 

Author Response

Response to Reviewer 2 Round 2 Comments

Page 2

Point 1. , not .

Response 1. Agreed. We changed the . by a ,

Point 2. Please check the order of the references - no 75 should probably be 12 (since it first appears in the Introduction). Probably you added them manually and not via a software like Mendeley or Zotero.

Response 2. Agreed. We changed the order of all references to meet the journal criteria as suggested by the reviewer.

Point 3. Please tell us more about the BAF biogeography. The Introduction section still seems rather short and abrupt. You could take inspiration from the following paper regarding the BAF details: Leão, Tarciso CC, Eimear Nic Lughadha, and Peter B. Reich. "Evolutionary patterns in the geographic range size of Atlantic Forest plants." Ecography (2020).

Response 3. Agreed. We rewrote the entire last paragraph from the introduction section, splitting it in two paragraphs providing more information about BAFs biogeography, as suggested.

“Previous phylogenetic studies for Malpighiaceae suggested a Brazilian Atlantic Rainforest (herein treated as BAF) origin for Stigmaphyllon, with S. paralias A.Juss. being consistently recovered in several studies as the first lineage to diverge in the genus [6,2]. Even though the BAFs currently comprises more than 15,000 known vascular plant species [12], they are regarded as one of the most threatened hotspots for biodiversity worldwide due to being mostly fragmented and disturbed by most of Brazil’s human population and economic activity [13]. Several biogeographic hypotheses have been proposed to explain the origins of the great biodiversity of the BAFs, such as: 1. Miocene to Pleistocene forest corridors between BAFs and the Amazon rainforests via Cerrado's gallery forests and/or via the coastal region of Northeastern Brazil [14]; and 2. Pleistocene refugia hypothesis, which suggested that Pleistocene climatic fluctuations led to rainforest fragmentation and promoted divergence of lineages or species in isolated forest fragments or refugia [15]. Pleistocene diversifications in BAFs have been heavily criticized due to the lack of concordance with empirical phylogenetic data, as well as by the evidence that shifts in forest species distribution, rather than fragmentation, have been the main consequences of global glaciations in the Neotropics [16]. Although most previous studies have focused on explaining BAFs biodiversity through Miocene to Pleistocene climatic/geological events, no study to date has focused on timing the age of vascular plant lineages diversification in BAFs.   

In this study, we focus on timing the biogeography history of the BAFs using Stigmaphyllon as a model genus, supplemented by a meta-analysis of vascular plant lineages diversified in this biome to infer the age of BAFs. More specifically, we: (1) test the monophyly of Stigmaphyllon and its subgenera; (2) time-calibrate the phylogenetic tree; (3) estimate the ancestral areas of Stigmaphyllon; and (4) shed some light on the age and biogeographic history of BAFs using Stigmaphyllon and a meta-analysis of vascular plant lineages diversified in this biome.”

Point 4. BAFs.

Response 4. Agreed. We changed the wording from BAFS to BAFs.

Point 5. model genus.

Response 5. Agreed. We rephrased to model genus, as suggested.

Point 6. supplemented by a.

Response 6. Agreed. We rephrased to supplemented by a, as suggested.

Point 7. Please also explain why a meta-analysis in needed. Please provide more details regarding the need of such an analysis. For instance, which genera have been analysed before? Is there a debate regarding the age of BAF? Why? What do we hope to gain from this?

Response 7. Agreed. We rewrote the entire third paragraph in the introduction section. Please, see response 3 for more details. 

Point 8. Infer.

Response 8. Agreed. We changed the verb from hypothesize to infer, as suggested.

Page 8

Point 9. Please at least consider adding a wider black frame for these white circles, because they are barely distinguishable.

Response 9. Agreed. We changed the colour of the white circles to green for better visualization.

Page 9

Point 10. If this stands for Atlantic Forest, then please clarify that in the legend.

Response 10. Agreed. We changed the phrasing to BAFs to standardize the text.

Page 15

Point 11. Do these 18 species account for a large proportion of the genus representatives in the BAF?

Response 11. Yes, there are 31 species of Stigmaphyllon in the BAFs, so it accounts for 60% of the genus diversity in the BAFs. We added the phrase [out of 31 spp.] in the text for a better understanding of the reader.

Page 16

Point 12. What do these three acronyms stand for? Please explain.

Response 12. Agreed. We added the full names for these acronyms in the captions.

Page 18

Point 13. Please state here the reasons why you did not use data from OA web databases such as GBIF or BIEN (i.e., your response to my comment in the previous review round).

Response 13. Agreed. We added the response suggested by the reviewer: “Online repositories such as GBIF, BIEN and Species Link were not used since specimen identification is not usually updated or are not identified by a Malpighiaceae specialist. Additionally, the main problem in using all the above-mentioned repositories is that only Flora do Brasil present reliable information on the biome distribution of species sampled in our study.”

Reviewer 3 Report

The manuscript is much clearer now and better structured. It can be published as it is.

Author Response

Thank you, so much, for all comments on the manuscript.

Round 3

Reviewer 2 Report

Dear authors,

Thank you for addressing all the issues raised during the review process. I have nothing more to add, except that you should consider stating in Table 2 that BAF stands for Brazilian Atlantic Forest.